# Modulatory Roles of ATP and Adenosine in Cholinergic Neuromuscular Transmission

**DOI:** 10.3390/ijms21176423

**Published:** 2020-09-03

**Authors:** Ayrat U. Ziganshin, Adel E. Khairullin, Charles H. V. Hoyle, Sergey N. Grishin

**Affiliations:** 1Department of Pharmacology, Kazan State Medical University, 49 Butlerov Street, 420012 Kazan, Russia; c.h.v.hoyle@gmail.com; 2Department of Biochemistry, Laboratory and Clinical Diagnostics, Kazan State Medical University, 49 Butlerov Street, 420012 Kazan, Russia; khajrulli@yandex.ru; 3Department of Medical and Biological Physics with Computer Science and Medical Equipment, Kazan State Medical University, 49 Butlerov Street, 420012 Kazan, Russia; sgrishin@inbox.ru

**Keywords:** neuromuscular junction, ATP, adenosine, purines, synaptic modulation

## Abstract

A review of the data on the modulatory action of adenosine 5’-triphosphate (ATP), the main co-transmitter with acetylcholine, and adenosine, the final ATP metabolite in the synaptic cleft, on neuromuscular transmission is presented. The effects of these endogenous modulators on pre- and post-synaptic processes are discussed. The contribution of purines to the processes of quantal and non-quantal secretion of acetylcholine into the synaptic cleft, as well as the influence of the postsynaptic effects of ATP and adenosine on the functioning of cholinergic receptors, are evaluated. As usual, the P2-receptor-mediated influence is minimal under physiological conditions, but it becomes very important in some pathophysiological situations such as hypothermia, stress, or ischemia. There are some data demonstrating the same in neuromuscular transmission. It is suggested that the role of endogenous purines is primarily to provide a safety factor for the efficiency of cholinergic neuromuscular transmission.

## 1. Introduction

The neuromuscular junction of vertebrates is the most well-studied cholinergic synapse. Acetylcholine (ACh) is the main neurotransmitter of the neuromotor unit of the somatic nervous system [1,2,3]. However, embryonic vertebrate skeletal muscle cells initially express receptors for adenosine 5′-triphosphate (ATP), adenosine, glutamate, γ-amino butyric acid (GABA), and glycine, as well as for ACh [4]. In the process of development, the expression of receptors for ACh begins to prevail [5,6] although purinoceptor-mediated synaptic modulation remains functionally significant [7,8].

The purinergic signaling system is represented by purine and pyrimidine nucleotides and nucleosides, which exert effects through families of adenosine, P2X and P2Y receptors [9]. The hypothesis of purinergic neural transmission was first put forward by G. Burnstock in the early 70s of the last century [10], and a little later, the presence of two types of receptors was suggested: P1 and P2 (for adenosine and ATP, respectively) [11]. This classification was subsequently revised, principally with the P1 receptor becoming the adenosine (A) receptor. Currently, A_1_, A_2A_, A_2B_, and A_3_ subtypes of adenosine receptors are distinguished. After several types of purine receptors had been cloned, it was suggested that ATP receptors should be divided into two families based on their molecular structure and activation mechanism: P2X receptors are ligand-gated ion channels and P2Y receptors are G protein-coupled receptors. Currently, this classification has been universally approved and to date, seven subtypes of P2X- (P2X1–7) and eight subtypes of P2Y receptors (P2Y_1,2,4,6,11-14_) have been described [12,13]. Identification of subtypes of purine receptors involved and their pharmacological characteristics are the initial stages of the study of the effector mechanisms of synaptic regulation by purines in the functioning of various skeletal muscles.

The synaptic modulatory function of the ATP molecule was discovered only at the end of the last century, and initially it was assigned a value only as a precursor of synaptic cleft adenosine [14,15,16]. Attempts to prove the synaptic effects of ATP were hindered by the generally recognized fact of the rapid destruction of ATP by ectonucleotidases to smaller nucleotides and ultimately to adenosine [17]. The first evidences of the participation of ATP in the processes of excitation in the central nervous system (CNS) was published in early 90s of the last century [18].

Further studies on embryonic and developing synapses have shown that ATP, as well as adenosine, modulates synaptic transmission [19,20]. It has now been established that ATP, without being degraded, can actively regulate the effectiveness of neuromuscular transmission [16,21,22,23,24,25], modulating the quantal and non-quantal release of the main transmitter which is ACh.

ATP as a co-transmitter accompanies a number of classical neurotransmitters, such as ACh, GABA, glycine, and glutamate [26], and is released upon activation of the presynaptic membrane into the synaptic cleft from the vesicles [27] or through some other physiological mechanisms [28]. It was established that in the innervated frog sartorius muscle ATP breakdown begins immediately, with adenosine being detectable within minutes, although its peak appearance is at 90–120 min [17]. The regulation of ATP and adenosine degradation is considered to be a novel mechanism to regulate skeletal muscle activity [29] although this needs to be developed further.

## 2. Effects of ATP and Adenosine on Presynaptic Currents

Assessment of the contribution of purines as modulators of synaptic transmission should begin with an analysis of their influence on the parameters of calcium current as an intermediate link between depolarization of the nerve ending and secretion of the neurotransmitter.

It should be noted that ATP has long been denied the realization of its own effect, including on calcium current. For example, a study by Hamilton and Smith [30] on the purinergic and cholinergic inhibition of calcium currents in rat motor nerve terminals provided evidence that ATP at a concentration of 50 μM reduced calcium currents by a third. This effect disappeared in the presence of α, β-methylene-adenosine-5’-diphosphate, which prevents ATP hydrolysis. From this, the authors concluded that the inhibitory effect of ATP occurs indirectly, through the action of the final degradation product of ATP which is adenosine. In their experiments, adenosine at the same concentration suppressed the terminal calcium current almost to the same extent as ATP. This effect was also shown for the analogue of adenosine, L-phenylisopropyl-adenosine, and was blocked by 8-para-sulfophenyl-theophylline, which is a non-selective antagonist of adenosine receptors.

It is interesting that in the same work it was shown how acetylcholine and the anticholinesterase agent, methanesulfonyl fluoride, like adenosine, suppressed the calcium current in the rat motoneuron terminal. This effect was reproduced by muscarine and was blocked both by atropine and the selective M1 receptor antagonist—pirenzepine [30]. Later, in 1995, using a frog neuromuscular preparation it was established that both ATP and adenosine can stimulate Ca^2+^ release from the intracellular stores of perisynaptic Schwann cells. These glial cell responses were seen when the motoneuron was stimulated. Blockade of ATP receptors, but not blockade of adenosine receptors, inhibited the neurally evoked glial cell calcium signal, indicating that P2 receptors on the perisynaptic Schwann cells are activated by endogenous ATP that is released when the motoneuron is stimulated [31]. Recently it was established that α7nAChR controls tetanic-induced ACh spillover from the neuromuscular synapse by promoting adenosine outflow from perisynaptic Schwann cells via ENT1 transporters and retrograde activation of presynaptic A_1_ inhibitory receptors [32]. In 2005, studies were conducted on perineural calcium currents [33], proving a direct relationship between ATP and calcium influx, while the effect of adenosine was characterized as calcium-independent [34].

The perineurium is poor in calcium channels, but it is a fairly isolated space within which calcium currents from all nerve endings are summed together. Here, using potassium channel blockers, tetraethylammonium, and 3,4-diaminopyridine, it is possible to record additive calcium currents. In our experiments on a frog sartorius neuromuscular preparation, the application of 100 μM ATP reversibly inhibited calcium currents by 17% without changing the sodium component [33]. Uridine-5′-triphosphate (UTP), an agonist of some subtypes of P2Y receptors, had the same effect. The action of ATP on calcium currents is prevented by the non-selective P2 receptor antagonist suramin. Unlike ATP or UTP, adenosine at a concentration of 100 μM did not change the parameters of calcium current [33].

Recently, the subtype of the calcium channel has been identified, on which ATP acts in the motor nerve ending of the frog cutaneous pectoris muscle; it turned out to be the L-type [35]. This is the same type of calcium channel that may be opened after adenosine A_2A_ receptor activation in rat motoneurons [36].

To check the possible effect of purines on the presynaptic potassium current in the frog neuromuscular junction we conducted experiments to record this current with an extracellular electrode from the proximal part of the nerve ending. All postsynaptic responses were blocked by (+)-tubocurarine. In this case, the second peak of the electrogram, immediately following the sodium component and counter-directed to it, reflects the potassium current of the nerve ending. ATP application did not change either the sodium or potassium component of the presynaptic action potential. Similarly, adenosine did not have a significant effect on presynaptic currents [33].

Thus, ATP inhibits the incoming calcium current in the nerve endings of the frog, but not sodium or potassium currents.

## 3. Effects of ATP and Adenosine on the Secretion of Acetylcholine

When the action potential reaches the nerve ending, ACh is released from the terminal of the motor neuron in an amount determined by the composition of the quantum. A quantum of a neurotransmitter is the number of its molecules contained in one presynaptic vesicle [37]. In response to one nerve impulse, the contents of dozens of vesicles are normally released, and quantal secretion occurs [38,39]. In the absence of excitation of the nerve terminal, a spontaneous release of the contents of usually one presynaptic vesicle with a frequency of once or twice per second is observed which is named single-quantum secretion [40,41,42]. There is also a non-quantum secretion of the neurotransmitter using an as yet unclear mechanism for the release of single molecules of acetylcholine from the nerve terminal into the synaptic cleft [43]. The two latter types of secretion have traditionally been given little functional importance, but recently, opinions on this subject tend to change.

### 3.1. Effects of ATP and Adenosine on Induced Quantal Secretion

As already mentioned, for a long time the generally accepted point of view was that exogenous ATP modulates synaptic transmission as a result of cleavage to adenosine [14]. Indeed, exogenous adenosine reduces by one third the quantal composition in synapses in phasic frog muscles, such as the cutaneous pectoris muscle. In this regard, it was assumed that it is adenosine that causes inhibition of synaptic currents during repetitive stimulation of the nerve [14]. According to these assumptions, it was believed that ATP in the neuromuscular junction plays the role of only adenosine source.

In 1998, the distinct presynaptic modulation of neuromuscular transmission by ATP and adenosine in frog neuromuscular junction was shown [21]. Although it had previously been established that adenosine inhibits neuromuscular transmission [44,45], it had not been shown that ATP itself could. It has been established that ATP and its derivatives: ADP, AMP, and adenosine are effective regulators of transmitter secretion. All of them have an inhibitory effect on the secretion of neurotransmitters from the nerve endings of phasic motor units. Moreover, the inhibitory effects of ATP, AMP, and adenosine in equimolar concentrations generally do not significantly differ from each other [24]. By recording the end-plate currents of frog neuromuscular junctions it was found that in the presence of ATP at concentrations of 50–100 μM the ACh quantal composition decreased by almost a third. It was suggested that this effect of ATP is realized through metabotropic P2Y receptors on the nerve terminal. This is supported by the facts that: (a) an antagonist of P2X purinoceptors, pyridoxalphosphate-6-azophenyl-2′,4′-disulfonic acid (PPADS), did not affect the effects of ATP; (b) the amplitude and duration of the action potential of the motor nerve ending did not change with the application of ATP; and (c) the inhibitory effect of ATP did not depend on a pH shift to either acidic or alkaline sides [21]. It was also very important to establish the summation of the action of ATP and adenosine, because with the combined application of these agents the overall effect was the exact sum of the inhibitory effect of each.

In continuation of these studies, we found [24,33,46] that at the frog neuromuscular junction ATP inhibits the ACh release acting via P2Y_12_ receptors. Activation of this receptor initiates a biochemical cascade, passing through the G_i/o_ subtype of the G protein to phospholipase C and then to protein kinase C [24]. Further, as shown in Figure 1, the signal goes in two parallel ways: through the production of hydrogen peroxide [47] and through the activation of phospholipase A_2_, the production of arachidonic acid and through cyclooxygenase synthesis of prostaglandin E_2_, which blocks the calcium channels of the nerve terminal [48].

Another mechanism of inhibition of quantal release of ACh has adenosine known for its neuroprotective function in the central nervous system [49]. At the neuromuscular junction, adenosine activates two subtypes of adenosine receptors simultaneously: A_1_ and A_2A_. While the A_1_ receptor mediates inhibition of ACh release, the A_2A_ receptor facilitates ACh release [50,51]. The second receptor subtype was found to be sensitive to the selective agonist of A_2A_ receptors CGS21680 and barium ions, which have a facilitating effect on neuromuscular transmission [52]. Stimulation of ACh release by the agonists of adenosine A_2A_ receptors is a promising direction to develop new drugs for treatment of Myasthenia gravis [53] while antagonists of these receptors are considered to be potential anti-parkinsonian drugs [54].

It was found that the selective agonist of adenosine A_1_ receptors, CCPA, has a more pronounced inhibitory effect on the amplitude of end-plate currents than adenosine, while its effects are eliminated by blockade of A-type potassium channels. No known intracellular second messengers or ion channels other than A-type of potassium channels were found to be involved in the effects of adenosine at the neuromuscular junction in the frog [55].

It was demonstrated that the regulation of ACh release in neuromuscular synapse involves many players on presynaptic level, including purinergic P1 and P2 receptors, muscarinic receptors, as well as receptors for trophic factors [56,57,58,59]. Interplay of these and may be other factors provides the adaptive changes in the synapse to a given condition [57].

In the terminal of motor neurons of amphibian phasic motor units, intracellular signaling mediated by P2Y_12_ receptors involves more metabolic processes and is more effective than that caused by activation of adenosine receptors. It was found that the action of ATP, but not adenosine, is attenuated by the stress hormone cortisol, which allows the efficiency of neuromuscular transmission to be increased sharply under stressful conditions [60,61,62]. It was also shown that methylprednisolone increases the outflow of ATP from resting motor endplates which may involve activation of presynaptic facilitatory adenosine A_2A_ receptors and synaptic vesicle redistribution [63].

In recent years, there has been an expansion in the field of studies of the modulating effects of purines in neuromuscular junctions [64,65,66,67,68,69,70,71,72,73,74,75,76,77,78,79]. Along with studies on the refinement of the subtype of the presynaptic P2 receptor involved in modulation in “classical” preparations [64,65,69,70,73,74,75,77,78], studies are being conducted to cover a broader spectrum of types of motor units in different laboratory animals [67,68,71,72,76,79] (see Table 1).

The inhibitory presynaptic effect of ATP, similar to that previously found in frogs [14,21,44,64,69], has been also shown for the rat slow soleus muscle [76] (see Figure 2). Interestingly, that in the fast extensor digitorum longus (EDL) muscle (rat) the presynaptic modulatory action of ATP is also inhibitory [71], but its effect is half of that, observed in the soleus slow muscle (Figure 2). As for the rat diaphragm, which is a muscle of mixed composition, we observed the opposite, presynaptic potentiating effect of ATP [74], while others [44,45,77,78] reported the inhibitory action of this purine (Table 1).

In contrast, in mouse fast and slow muscle preparations the presynaptic action of ATP is potentiation [71,72], while in mouse diaphragm it is reported to be either potentiation [74] or inhibition [25,65,75] (Table 1).

Results presented in the Table 1 show some current uncertainty in our understanding of purine effects on various skeletal muscle types of various species due to inconsistency of reported effects. For example, there are some contradicting results in rodent (mouse and rat) diaphragm on presynaptic level. There may be multiple reasons for such inconsistencies, including stimulation parameters or other marked variations of experimental designs and protocols. Exploring the measure and direction of the impact of changing of each component of experimental design/protocols deserves a special separate research project. Given that the scope of the current manuscript is to provide a general overview of the effects of adenosine and ATP on neuromuscular transmission, we opted to keep the Table 1 readable and not to overload it with experimental protocol specifics but rather to secure this exciting work for the new future project.

### 3.2. Effects of ATP and Adenosine on Spontaneous Quantal Secretion

It is known that the inhibitory effect of cholinomimetics on the spontaneous quantal secretion of ACh into the neuromuscular junction of a frog is due to activation of presynaptic muscarinic M2 receptors [80]. Interestingly, in the presence of adenosine (which has a similar effect to cholinomimetics on quantal secretion), carbachol did not display its own inhibitory effect on the quantal secretion of ACh [81].

Muscarinic receptor antagonists, atropine and AFDX-116, do not affect the inhibitory effect of adenosine, and antagonists of the adenosine receptors, 8-SPT and DPCPX, do not modify the effect of carbachol [80]. These data prove the absence of direct membrane interactions between adenosine and muscarinic receptors. Both carbachol and adenosine retain their inhibitory effects on enhanced single-quantum transmitter release evoked by ionomycin, sucrose, and a high potassium content [81]. According to the authors, there is a mechanism that causes the occlusion of the metabolic pathways following activation of the adenosine and muscarinic receptors [81].

ATP, like adenosine, does not change the amplitude-time parameters of spontaneous end-plate responses, and its effect on reducing the frequency of these single-quantum responses is known, both in neuromuscular junctions of cold-blooded animals (frog) [21,24] as well as in warm-blooded ones (mouse) [23,65,66]. The subtype of P2 receptor involved in this modulating presynaptic inhibitory action of ATP at neuromuscular junctions in the diaphragm of mice has been specifically determined to be the P2Y_13_ subtype in experiments evaluating spontaneous secretion [65]. Initially, the authors confirmed that the selective P2Y_12–13_ receptor agonist, 2-methylthio-ADP (2-MeSADP), reduces the frequency of miniature end-plate potentials without affecting their amplitude. The effect on spontaneous secretion disappeared after the application of selective antagonists of P2Y_12-13_ receptors, AR-C69931MX and 2-MeSAMP. 2-MeSADP was more effective than ADP and ATP in reducing the frequency of spontaneous end-plate potentials. It was then demonstrated that the selective antagonist of P2Y_13_-receptors, MRS-2211, abolished the inhibitory effect of 2-MeSADP on the frequency of single-quantum responses, whereas the antagonist of P2Y_12_ receptors, MRS-2395, did not. The selective agonist of P2Y_13_ receptors, inosine-5’-diphospate (IDP), reduced spontaneous secretion of acetylcholine, and MRS-2211 abolished IDP-mediated modulation [65].

The authors continued their studies on the same preparation in order to elucidate the effect of purines on the K^+^-dependent release of ACh [66]. They found that when the motor nerve endings were depolarized by an increased concentration of K^+^, ATP and endogenously applied adenosine were able to modulate the secretion of ACh by sequential activation of various purinergic receptors.

### 3.3. Effects of ATP and Adenosine on Non-Quantal Secretion

It is known that at rest almost all release of ACh from the nerve endings into the synaptic cleft occurs in a non-quantal manner [82,83]. In a typical experiment, a whole muscle preparation is incubated in the presence of an acetylcholinesterase blocker and then treated with (+)-tubocurarine. The recorded hyperpolarization is called the H-effect obtained by elimination of non-quantal acetylcholine depolarization [84]. Non-quantal secretion, which is two orders of magnitude higher than quantal (at rest), is traditionally given less importance in the analysis in view of its tonic non-cumulative effect. Meanwhile, a decrease in the level of membrane potential by ~5 mV under the action of non-quantal ACh significantly affects the activity of the postsynaptic membrane, and, in particular, the sensitivity of cholinergic receptors.

We have shown that, in mouse diaphragm preparations, ATP reduces the H-effect from 5 to 1.5 mV [23]. This could happen when the input membrane resistance or sensitivity to acetylcholine changes, but we verified that these did not change. Adenosine did not affect non-quantal release. Inhibitors of protein kinase A or guanylate cyclases, Rp-cAMP and ODQ respectively, did not affect the inhibition of the H-effect by ATP. However, staurosporine, a protein kinase C inhibitor, abolished the effect of ATP. Suramin, a non-selective P2 receptor antagonist, enhanced the H-effect to 8.6 mV and, like staurosporine, prevented the inhibitory effect of ATP. We concluded that ATP inhibits non-quantal release by directly affecting the presynaptic metabotropic P2 receptors associated with protein kinase C. These data show that non-quantal release of acetylcholine can be modulated by endogenous ATP [23].

Experiments to evaluate the effect of purines on non-quantal secretion at neuromuscular junctions were continued by studies on rat preparations [83]. ATP at a concentration of 100 μM reduced the H-effect at neuromuscular junctions in the rat diaphragm by 66%. This effect of ATP remained unchanged after blocking P2X receptors by Evans blue and PPADS, but was abolished by the non-selective P2 antagonist, suramin, Reactive blue 2 (an antagonist of some metabotropic P2 receptors), as well as U73122 (a phospholipase C inhibitor).

The action of ATP is not associated with voltage-dependent Ca^2+^ channels. With the simultaneous use of ATP and glutamate, the additive depression led to the complete disappearance of the H-effect. This can be explained by the independence of the action of ATP and glutamate [83].

Adenosine did not affect the H-effect in the rat diaphragm in any way [83]. Thus, we are faced with a fundamental difference in the effect of the two purines that we have analyzed on the level of non-quantal secretion.

We can assume that ATP, reducing the non-quantal output of the ACh in the synapses of the respiratory muscles, reduces the number of desensitized postsynaptic receptors, thereby increasing the number of activated cholinergic receptors in response to the quantal output of ACh. However, this postsynaptic effect should have been accompanied by an increase in the amplitude of spontaneous single-quantum responses, which, as already mentioned above, was not noted [23,65,66].

## 4. Post-Synaptic Effects of ATP and Adenosine

While at a presynaptic level ATP and its derivatives exert a predominantly inhibitory effect on neurotransmitter secretion in non-diaphragmatic muscles in rats, the postsynaptic effect of ATP on neuromuscular transmission is unpredictable [71,72,74,76]. The direction of its postsynaptic effect in fast muscles is opposite what is observed in other types of skeletal muscles. Thus, at neuromuscular junctions in rat and mouse EDL muscles, we have observed a postsynaptic inhibitory effect of ATP on neuromuscular transmission [71]. At the synapses of the slow soleus muscle, there was no significant postsynaptic effect of ATP at physiological temperature [72,76]. Interestingly, with hypothermia, the postsynaptic facilitation effect of ATP becomes more pronounced than the presynaptic one (Figure 2). We have not observed any pronounced postsynaptic action for adenosine [71,76].

Recently, a correlation between P2 signaling and acetylcholinesterase activity has been found [85]. Acetylcholinesterase is found in a large quantity in the synaptic cleft at the motor end-plate [86]. It immediately hydrolyses ACh into acetate ion and choline. Changes in its activity directly affect the activity of postsynaptic cholinergic receptors. In mice with a P2Y_1_ receptor knock-out the expression of acetylcholinesterase in the muscles is significantly reduced [85].

Purinergic signaling is known to be extremely thermosensitive [87,88] and the postsynaptic effect of ATP is weakly expressed under normothermic conditions [71,72,74,76]. We found that in rat soleus muscle the presynaptic inhibitory effect of exogenous ATP decreased exponentially until it completely disappeared at 14 °C, while at postsynaptic level ATP had no effect at normal temperature, but at low temperature, it potentiated the carbachol-induced contractions [76]. In rat EDL muscle, there were no such striking difference of ATP effects on pre- and postsynaptic levels at different temperatures [71].

It is known that the potentiating effect of ATP is manifested in embryonic and developing muscles [89]. Adenosine increases ε-nAChR-channel activity at the end-plate in adult skeletal muscle fibers where the interplay between adenosine and cholinergic systems is mainly mediated by A_2B_ receptors and partially by A_3_ receptors [90]. ATP increases the time of the open state and the frequency of opening of acetylcholine-activated ion channels in embryonic muscles [19,91].

It is possible that ATP is able to act directly on nicotinic cholinoceptors. This has been noted in mature rat muscles [20] and in cultured frog muscle cells [92]. It has been established that the nicotinic receptor has ATP binding sites [93]. Based on these data, it can be assumed that the facilitating effect of endogenous ATP, secreted in conjunction with acetylcholine as a cotransmitter, can modulate neuromuscular transmission at the postsynaptic level.

The phenomenon of postsynaptic potentiation acts as the most significant process in which ATP could be involved [94,95]. In particular, it seems that ATP secreted together with ACh forms a trace reaction stored on the postsynaptic membrane after a single signal for up to tens of seconds. This is manifested in a slowing down of the decay rate of the second current of the end-plate during paired stimulation of the motor nerve, as well as an increase in its amplitude. This is supported by the observation that exogenous ATP almost eliminates the phenomenon of an increase in the duration of paired synaptic currents [95].

With inhibited acetylcholinesterase, a decrease in the decay of a single current of the end-plate occurs which indicates a postsynaptic potentiating effect. In this situation exogenous ATP acts on current decay similarly to exogenous ACh which can also slow down synaptic signal decay [96]. At the same time, we must not forget the fact that, with active acetylcholinesterase, neither ATP [21,23,24] nor acetylcholine [97] affect the time course and potential-dependence of the decrease in miniature end-plate currents.

Currently it is difficult to give an unambiguous explanation for the effect of reducing postsynaptic potentiation by ATP. It is attractive to assume that endogenous ATP released after the first nerve impulse remains for seconds in the synaptic cleft potentiating responses to ACh during the generation of a subsequent end-plate current [96]. However, the situation is complicated by the presynaptic action of ATP.

Another site of action of ATP is associated with the specific location of P2X4 receptors on the membranes of T-tubules, and suggests their contribution to the mechanism of excitation–contraction coupling. Activation of P2X4 receptors leads to the entry of calcium ions into the muscle cell, which may contribute to the re-filling of the sarcoplasmic reticulum or activation of Ca^2+^-dependent signaling pathways [97].

P2X7 receptors are localized in different regions of the neuromuscular junction [98,99]. To activate the P2X7 receptors the concentration of ATP needs to be much higher (>100 μM) than for other P2X receptors [100]. During tissue ischemia, ATP is released at high concentration from the cells and activates P2X receptors [101]. Since the P2X7 receptor does not desensitize, the ion channel remains open as long as the receptor is bound to ATP, and activation of the P2X7 receptor leads to a strong disruption of the ionic equilibrium of the cytoplasm [102]. Thus, it has been suggested that P2X receptors can be targets for the treatment of dystrophic diseases. In the early stages of dystrophic muscle development Duchenne muscular dystrophy, when calcium homeostasis is impaired, P2X receptors (possibly P2X7, P2X4 and P2X5) can significantly contribute to the pathological entry of Ca^2+^ into the muscle [100].

## 5. Conclusions

ATP acting on P2 receptors modulates quantal and non-quantal secretion of the neurotransmitter, ACh, from the presynaptic terminals of motor neurons. The end product of its breakdown, adenosine, realizes its action through adenosine receptors and also modulates the release of ACh. ATP, in addition, can affect the level of non-quantal secretion and the state of activation of postsynaptic cholinergic receptors.

Thus, the appearance of endogenous ATP in the synaptic cleft of the neuromuscular junction in an effective concentration at which it can have both presynaptic and postsynaptic effects suggests that this purinergic agent is able to function as an endogenous modulator of synaptic transmission, regulating the secretion of the primary transmitter, ACh, and increasing the efficiency of transduction mechanisms in the postsynaptic membrane. This combination of pre- and postsynaptic effects of ATP can be considered as one of those safety factors mentioned by [103], when with a smaller amount of released ACh a greater reliability of neuromuscular transmission can be maintained. With the accumulation of large concentrations of ATP and the product of its hydrolysis of adenosine, one would expect an increase in the synaptic effects of endogenous purines. These data open up new prospects for targeted pharmacological regulation of quantal secretion through this receptor mechanism.

The article focused on the phasic muscles, and one should not forget about the various effects of purines in tonic motor units [104], data about which were only touched on in this review and require separate publication. As for the predominance of attention in the article to one purine, ATP, this is primarily to correct historical injustice, when for decades its own synaptic effects have not been fully recognized, giving exceptional preference to its metabolite—adenosine [105].

## Figures and Tables

**Figure 1 ijms-21-06423-f001:**
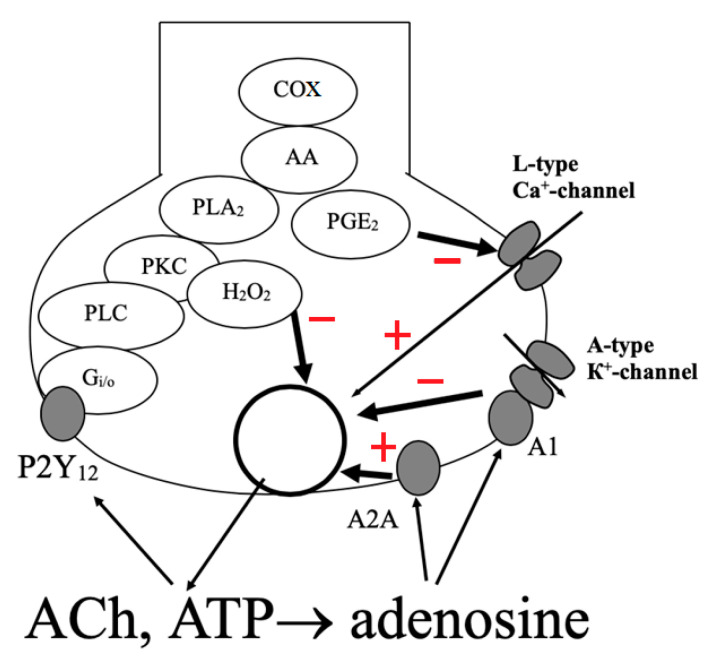
Schematic diagram of presynaptic regulation by ATP of neuromuscular transmission in frog phasic muscles. The synaptic vesicle filled with ACh and ATP releases its content by exocytosis into the synaptic cleft. ATP and its metabolite, adenosine, are both involved in separate negative feedback mechanisms: ATP via P2Y_12_ receptors and adenosine via adenosine A_1_ and A_2A_ receptors. AA—arachidonic acid; PGE_2_—prostaglandin E_2_; PKC—protein kinase C; PLA_2_—phospholipase A_2_; PLC—phospholipase C; COX—cyclooxygenase; H_2_O_2_—hydrogen peroxide.

**Figure 2 ijms-21-06423-f002:**
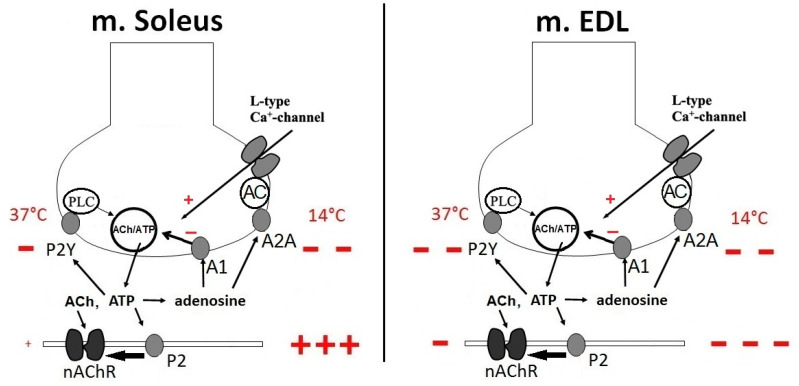
Schematic diagram of purinergic modulation of neuromuscular transmission at soleus muscle (m. Soleus) and extensor digitorum longus (m. EDL) neuromuscular junctions under normothermic and hypothermic conditions. ATP and its metabolite, adenosine, are both involved in separate negative feedback mechanisms: ATP via P2Y receptors and adenosine via adenosine A_1_ and A_2A_ receptors. A synaptic vesicle in the nerve terminal is shown, filled with ATP in addition to the primary neurotransmitter, acetylcholine (ACh). Exocytosis of the vesicle contents causes activation of ATP pre- and post-synaptic P2 receptors. At 14 °C the pre- and post-junctional effects become more pronounced. The signs “+” and “−” show the direction of action of ATP: +, facilitating neuromuscular transmission, −, inhibiting neuromuscular transmission. Doubling and tripling of signs means, respectively, two- and three-fold more pronounced effects. The size of the signs themselves also indicates a relative magnitude of the effects. PLC—phospholipase C; AC—adenylate cyclase.

**Table 1 ijms-21-06423-t001:** Effects of adenosine and ATP at neuromuscular junctions of various types of skeletal muscle.

Animal	Muscle Name	Muscle Type	Synaptic Level	Frequency of Electrical Stimulation (Hz)	Agent		References
					Adenosine	ATP	
Crayfish	Walking leg	-	Pre-synaptic	0.5–2	NE	↓	[67]
				-	NE	NT	[68]
Frog	Sartorius	-	Pre-synaptic	0.05	NT	↓	[14]
				0.03	↓	↓	[21]
				0.5	NT	↓	[44]
				0.03	NT	↓	[64]
				0.5	↓	NT	[69]
Guinea pig	Diaphragm	Mixed	-		NE	NT	[70]
Mouse	EDL	Fast	Pre-synaptic	0.1	↓	↑	[71]
			Post-synaptic	-	NE	↓	[71]
	Soleus	Slow	Pre-synaptic	0.1	NT	↑	[72]
			Post-synaptic	-	NT	NE	[72]
	Diaphragm	Mixed	Pre-synaptic	0.5	NT	↓	[65]
				0.5	↓	NT	[73]
				0.1	NT	↑	[74]
				1	NE	↓	[75]
			Post-synaptic	-	NT	↑	[74]
Rat	EDL	Fast	Pre-synaptic	0.1	↓	↓	[72]
			Post-synaptic	-	NE	↓	[72]
	Soleus	Slow	Pre-synaptic	0.1	↓	↓	[76]
			Post-synaptic	-	NE	NE	[76]
	Diaphragm	Mixed	Pre-synaptic	0.5	NT	↓	[44]
				0.5	↓	NT	[77]
				0.1	↓ or ↑ *	↓	[78]
				0.1	NT	↑	[74]
			Post-synaptic	-	NT	↑	[74]
	LAL	Fast	Pre-synaptic	2	NT	↑	[79]

Notes: The arrows indicate the direction of the effect: ↑—facilitation; ↓—inhibition. EDL—extensor digitorum longus; LAL—levator auris longus; NE—no effects; NT—not tested; *—depends on concentration.

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
