# Peer review of "Modulatory Roles of ATP and Adenosine in Cholinergic Neuromuscular Transmission"

_ijms, 2020, doi:10.3390/ijms21176423_

Round 1

Reviewer 1 Report

This manuscript reviews the modulatory action of ATP and Adenosine on neuromuscular transmission. Indeed other reviews have been published on similar topics in recent years (e.g. Bernareggi et al, 2019). However the topic  remains interesting and the manuscript is exhaustive. Also the conclusion that the role of endogenous purines is primarily to provide a safety factor for the efficiency of cholinergic neuromuscular transmission deserves interest.

In my opinion, the efficacy of Figures 2 and 3 would increase if they were unfied

Author Response

Dear Reviewer,

Thank you for your positive comments on our paper.

This manuscript reviews the modulatory action of ATP and Adenosine on neuromuscular transmission. Indeed other reviews have been published on similar topics in recent years (e.g. Bernareggi et al, 2019). However the topic  remains interesting and the manuscript is exhaustive. Also the conclusion that the role of endogenous purines is primarily to provide a safety factor for the efficiency of cholinergic neuromuscular transmission deserves interest.

Our response:

The review of Bernareggi et al., 2019 [54] concerns interaction of adenosinergic and cholinergic transmission. In this our paper we try to shift focus on “ATP-ergic” processes and only touching adenosinergic.

In my opinion, the efficacy of Figures 2 and 3 would increase if they were unfied

Our response:

Thank you for this suggestion. We have unified the Figures 2 and 3 - now it is Figure 2.

Reviewer 2 Report

The topic of the review is certainly of interest. A review is expected to be unbiased and to respect the contribution of the main researchers who were paramount to gather the key concepts of the topic being reviewed. This is simply not respected in this review, where the eagerness for self-citation far exceeds the care to be right and fair to give credit to the pioneering studies or to views different from these developed by the authors. Too often the contributions of the founding fathers are forgotten and their original contributions replaced by citations to more recent works, several of them largely confirmatory or with minor conceptual advances.

-‘The synaptic modulatory function of the macroergic ATP molecule was discovered only at the end of the last century’: The proposal that extracellular ATP may act as a signaling molecule was first proposed by Szent Gyorgy in 1921.

-[ATP]’it was assigned a value only as a precursor of synaptic cleft 42 adenosine [13]’: It is simply outrageous for this reviewer to see history being re-written by newcomers to the field. The interest of purines as modulators at neuromuscular junctions began in Ginsborg's lab, with the work of Alex Ribeiro and Gene Silinsky (Ribeiro & Walker J, 1973, Br J Pharmacol 49: 724; Silinsky, 1975, J. Physiol. 247. 145). Unless the work of these two giants is dully acknowledged and EXTENSIVELY quoted and acknowledge as pioneering, I will make all efforts to ecommemnd refusing the publication of this review.

-‘the rapid destruction of ATP by ectonucleotidases to smaller nucleotides and ultimately to adenosine [14]’: Another unacceptable quotation! Ecto-nucleotidases at the neuromuscular juntion were first characterized by Cunha & Sebastiao, 1991, Eur J Pharmarol 197: 83), all other works were only confirmatory.

-‘The first evidences of the participation of ATP in the processes of 45 excitation in the CNS was published in early 90-th of the last century [15, 16]’: These references are WRONG! The first quoted paper is in a ganglion, the second an editorial. The first demonstration of ATP as neurotransmitter in the CNS was provided by Francis Edwards (Edwards et al., 1992, Nature 359: 144-7).

-lines 47-50: The choice of these references [17-23] is only about ATP and Silinskly's work are missing (any of Silinsky's references could replace the authors' self-referencing with increased quality, novelty and property). Either remove mentioning adenosine or insert the numerous works of Ribeiro's group, which seem to have been forgotten. This unacceptable bias needs to be corrected.

-‘ATP as a cotransmitter accompanies a number of classical neurotransmitters, such as ACh, GABA, glycine and glutamate [24]’: Quoting this obscure paper seems like a joke in view of the numerous and far superior reviews by leaders in the area on the topic of ATP release. This is the sort of bias that can only be ranked as unacceptable.

-lines 61-70: The impact of ATP and its P2R on calcium at NMJ was superiorly investigated by Robitaille's group, after the pioneering work of Silinsky's group. The impact of adenosine on calcium channels involves A1 R (Silinsky's work) and A2AR (Correia-de-Sa's work). A lot (too much) is missing in the eagerness to jump to self quotation.

-lines 104-113: Even the work of Nobel prize winners are forgotten! Sir Bernard Katz must be rolling and jumping in his coffin with the way his contributions are ignored. Re-organize all this introduction to point 3. after reading (and hopefully understanding the absolute pioneering and still fully actual novelty) of Katz's work.

After such a display of mis-quotations that seem like an attempt to bend the literature to the sole interests of the authors, this Reviewer (as any informed reader) will be highly suspicious. Therefore, I just read the rest of the text en passant and further notes can be issued only, and only if, all the above mentioned correction and completely resolved.

Other general points:

-Figure 1 seems to imply that a lot more is known about the transducing systems linking P2R to ACh release when compared to the lack of detail of adenosine. I am afraid that an objective evaluation of the literature will allow concluding that adenosine modulation at the NMJ is more evident and with greater impact in different physiopathological conditions and far more is known about how adenosine receptors orchestrate cholinergic and peptidergic auto-modulation to adjust transmission to the pattern of recruitment of these NMJs. Either isolate ATP effects in this Figure or complete with what is known for adenosine.

- Figures 2 and 3 are nearly identical. No such duplication is required. P1 as A1R is showing a lack of understanding of P1R (see Correia-de-Sa extensive work).

-transmission the purinergic influence is minimal under physiological conditions: This might be acceptable for P2R-mediated effects, but it is certainly not acceptable for A1R/A2AR as beautifully illustrated by the work of Correia-de-Sá.

-‘the macroergic ATP molecule’: Phosphodiester bonds are 'low energy' compared to carbon-carbon bonds: in terms of energy, the latter are like 100 dollar bills and the former like 1 dollar bills. Not surprisingly, biology selected an easy manageable currency for most purposes since the use of high currency would entail either big losses (no change) or too much heat produced locally. Therefore, it is not correct to consider ATP as a high energy molecule,; it is simply an easy and plastic energy transfer system.

Author Response

Responses to the Reviewer 2.

  1. The topic of the review is certainly of interest. A review is expected to be unbiased and to respect the contribution of the main researchers who were paramount to gather the key concepts of the topic being reviewed. This is simply not respected in this review, where the eagerness for self-citation far exceeds the care to be right and fair to give credit to the pioneering studies or to views different from these developed by the authors. Too often the contributions of the founding fathers are forgotten and their original contributions replaced by citations to more recent works, several of them largely confirmatory or with minor conceptual advances.

Our response:

It is not always necessary to go back to first principles.  For example, no one normally cites the work of Dale when writing about cholinergic motor transmission.  This review is not intended to be an historical review, thus, to our mind it is not necessary to cite the early papers of nearly one hundred years ago on ATP and related compounds as signal molecules.  The field has moved on from requiring that sort of justification as it is now well accepted that these compounds act as signal molecules in a variety of ways.

Also, the requirement of the Editor for this review was that “References should be up-to-date, i.e., 50% or above are the papers published within recent 5 years” which we tried to meet.

  1. –‘The synaptic modulatory function of the macroergic ATP molecule was discovered only at the end of the last century’: The proposal that extracellular ATP may act as a signaling molecule was first proposed by Szent Gyorgy in 1921.

Our response:

We are not familiar with Szent Gyorgi (1921), however, in Drury, A.N. and Szent-Györgyi,A.(1929). The physiological activity of adenine compounds with especial reference to their action upon the mammalian heart. J. Physiol., 68, 213-237, pharmacological effects of AMP and adenosine on the cardiovascular system were reported.  Although this is undoubtably a seminal paper in the purines field, it is not directly relevant to the actions of ATP that we are concerned with here.

  1. -[ATP]’it was assigned a value only as a precursor of synaptic cleft 42 adenosine [13]’: It is simply outrageous for this reviewer to see history being re-written by newcomers to the field. The interest of purines as modulators at neuromuscular junctions began in Ginsborg's lab, with the work of Alex Ribeiro and Gene Silinsky (Ribeiro & Walker J, 1973, Br J Pharmacol 49: 724; Silinsky, 1975, J. Physiol. 247. 145). Unless the work of these two giants is dully acknowledged and EXTENSIVELY quoted and acknowledge as pioneering, I will make all efforts to ecommemnd refusing the publication of this review.

Our response:

  1. Silinsky, E.M.; Hunt, J.M.; Solsona, C.S.; Hirsh, J.K. Prejunctional adenosine and ATP receptors. NY Acad. Sci. 1990, 603, 324-333. doi: 10.1111/j.1749-6632.1990.tb37683.x
  2. Silinsky, E.M.; Hirsh, J.K.; Searl, T.J.; Redman, R.S.; Watanabe, M. Quantal ATP release from motor nerve endings and its role in neurally mediated depression. Brain Res. 1999, 120, 145-158. doi: 10.1016/s0079-6123(08)63552-9
  3. Ribeiro, J.A.; Walker, J. Action of adenosine triphosphate on endplate potentials recorded from muscle fibres of the rat-diaphragm and frog sartorius. J. Pharmacol. 1973, 49, 724-725.
  4. Ribeiro, J.A.; Dominguez, M.L. Mechanisms of depression of neuromuscular transmission by ATP and adenosine. Physiol. (Paris), 1978, 74, 491-496.

  1. -‘the rapid destruction of ATP by ectonucleotidases to smaller nucleotides and ultimately to adenosine [14]’: Another unacceptable quotation! Ecto-nucleotidases at the neuromuscular juntion were first characterized by Cunha & Sebastiao, 1991, Eur J Pharmarol 197: 83), all other works were only confirmatory.

Our response:

This is an error, we do, in fact refer to Cunha and Sebastiao, later in the manuscript, but we should have cited this work here, and now do so (see lines 44-45, and 53-55)

  1. Cunha, R. A.; Sebastião, A. M. Extracellular metabolism of adenine nucleotides and adenosine in the innervated skeletal muscle of the frog. J. Pharmacol. 1991, 197, 83-92. doi:10.1016/0014-2999(91)90368-z.

  1. -‘The first evidences of the participation of ATP in the processes of 45 excitation in the CNS was published in early 90-th of the last century [15, 16]’: These references are WRONG! The first quoted paper is in a ganglion, the second an editorial. The first demonstration of ATP as neurotransmitter in the CNS was provided by Francis Edwards (Edwards et al., 1992, Nature 359: 144-7).

Our response:

Thank you for picking up this error. [15] concerns a sympathetic ganglion, and is therefore incorrect here.  [16] is also incorrect, because it is a commentary in Nature on the paper by Edwards et al.

We now cite here (see lines 45-46)

  1. Edwards, F.A.; Gibb, A.J.; Colquhoun, D. ATP receptor-mediated synaptic currents in the central nervous system. Nature. 1992, 359, 144-147. doi: 10.1038/359144a0.

  1. -lines 47-50: The choice of these references [17-23] is only about ATP and Silinskly's work are missing (any of Silinsky's references could replace the authors' self-referencing with increased quality, novelty and property). Either remove mentioning adenosine or insert the numerous works of Ribeiro's group, which seem to have been forgotten. This unacceptable bias needs to be corrected.

Our response:

References 17 and 18 (now [18, 19]) concern embryonic and developing synapses, neither of which have been addressed by Silinsky.  References 20 to 24 concern ATP and not adenosine or any other degradation product of ATP; [21] and [24] are papers from Salgado et al. (Ribeiro's group) and De Lorenzo et al., respectively.  Silinsky showed that although ATP is co-released with ACh, it is broken down by ecto-enzymes and subsequent prejunctional inhibition is caused by adenosine, not ATP, at frog neuromuscular junctions [13].

We wish to clarify or meaning and have changed the sentence to read: "It has now been established that ATP, without being degraded, can actively…" (see lines 48-50).

  1. –‘ATP as a cotransmitter accompanies a number of classical neurotransmitters, such as Ach, GABA, glycine and glutamate [24]’: Quoting this obscure paper seems like a joke in view of the numerous and far superior reviews by leaders in the area on the topic of ATP release. This is the sort of bias that can only be ranked as unacceptable.

Our response:

We chose this article (now [25]) because it is recent and covers the ground that is pertinent as it covers cotransmission by motoneurones.  It is published in a reputable journal that is readily accessible, which has a reasonable impact factor.  In the same sentence we cite a recent review by Burnsock (2020) [26] and a review by Gundersen et al. (2015) [27].  In the following sentence we cite Cunha and Sebastião (1991) [16].

See lines 51-55.

  1. -lines 61-70: The impact of ATP and its P2R on calcium at NMJ was superiorly investigated by Robitaille's group, after the pioneering work of Silinsky's group. The impact of adenosine on calcium channels involves A1 R (Silinsky's work) and A2AR (Correia-de-Sa's work). A lot (too much) is missing in the eagerness to jump to self quotation.

Our response:

We cite Robitaille [29] (see lines 77-80). Silinksy has never found an ATP receptor at either amphibian or rodent neuromuscular junctions, although he did, indeed, describe the degradation of ATP to adenosine at these NMJs (Silinsky et al. Ann NY Acad Sci 1990;603:324-333, Silinsky et al. Prog Brain Res 1999;120:145-158), and an adenosine A1 receptor in the frog (Redman and Silinsky, Mol. Pharmacol. 1993; 44:835-840), he did not describe the effects of ATP on calcium channels or mobilisation.  We have cited two further papers by Silinsky in the revised Introduction, as mentioned above. Since we are more concerned with ATP acting per se it does not seem appropriate to cite Silinsky's papers here. However, we have added a sentence and now cite Correia-de-Sá:

  1. Correia-de-Sá, P.; Timóteo, M.A.; Ribeiro, J.A. A(2A) Adenosine receptor facilitation of neuromuscular transmission: influence of stimulus paradigm on calcium mobilization. J. Neurochem. 2000, 74 (6): 2462-2469 doi: 10.1046/j.1471-4159.2000.0742462.x.

  1. -lines 104-113: Even the work of Nobel prize winners are forgotten! Sir Bernard Katz must be rolling and jumping in his coffin with the way his contributions are ignored. Re-organize all this introduction to point 3. after reading (and hopefully understanding the absolute pioneering and still fully actual novelty) of Katz's work.

Our response:

We did not feel it necessary to re-tread well trodden ground, so we cited a recent (2018) review [39].  However, we have now included two citations, to Fatt and Katz (1952) and del Castillo and Katz (1955) (see lines 110-112)

  1. Fatt, P.; Katz, B. Spontaneous subthreshold activity at motor nerve endings. Physiol. 1952, 117, 109-128.
  2. del Castillo, J.; Katz, B. (1955) Local activity at a depolarized nerve-muscle junction. J. Physiol. 1955, 128, 396–411. doi: 10.1113/jphysiol.1955.sp005315

  1. After such a display of mis-quotations that seem like an attempt to bend the literature to the sole interests of the authors, this Reviewer (as any informed reader) will be highly suspicious. Therefore, I just read the rest of the text en passant and further notes can be issued only, and only if, all the above mentioned correction and completely resolved.

Our response:

We found such comment inappropriate and unprofessional. As you see from above, we have tried to extract from your review constructive suggestions and did our best to meet your requirements. However, we cannot accept such arrogant style of communication.

Other general points:

  1. -Figure 1 seems to imply that a lot more is known about the transducing systems linking P2R to ACh release when compared to the lack of detail of adenosine. I am afraid that an objective evaluation of the literature will allow concluding that adenosine modulation at the NMJ is more evident and with greater impact in different physiopathological conditions and far more is known about how adenosine receptors orchestrate cholinergic and peptidergic auto-modulation to adjust transmission to the pattern of recruitment of these NMJs. Either isolate ATP effects in this Figure or complete with what is known for adenosine.

Our response:

We change the legend of the Figure 1. (see lines 150-151).

Figure 1. Schematic diagram of presynaptic regulation by ATP of neuromuscular transmission in frog phasic muscles.

  1. - Figures 2 and 3 are nearly identical. No such duplication is required. P1 as A1R is showing a lack of understanding of P1R (see Correia-de-Sa extensive work).

Our response:

The figures are similar but not identical.  In two different muscles from the same animal, neuromodulations of cholinergic responses by ATP, mediated via P2 receptors, have opposite directions, as shown in the two figures.

However, since both Reviewers suggested that, we have unified the Figures 2 and 3 - now it is Figure 2.

  1. -transmission the purinergic influence is minimal under physiological conditions: This might be acceptable for P2R-mediated effects, but it is certainly not acceptable for A1R/A2AR as beautifully illustrated by the work of Correia-de-Sá.

Our response:

We change this to

“In neuromuscular transmission the P2-receptor-mediated influence is minimal under physiological conditions but it becomes very important in some pathophysiological situations such as hypothermia, stress or ischemia.” (see lines 14-16)

  1. -‘the macroergic ATP molecule’: Phosphodiester bonds are 'low energy' compared to carbon-carbon bonds: in terms of energy, the latter are like 100 dollar bills and the former like 1 dollar bills. Not surprisingly, biology selected an easy manageable currency for most purposes since the use of high currency would entail either big losses (no change) or too much heat produced locally. Therefore, it is not correct to consider ATP as a high energy molecule,; it is simply an easy and plastic energy transfer system.

Our response:

Thank you for this excellent analogy.  We have deleted "macroergic" (see line 41).

Reviewer 3 Report

IJMS-868123 - Modulatory Roles of ATP and Adenosine in Cholinergic Neuromuscular Transmission

This is a revised version of Ziganshin et al manuscript on the role of purines, mainly ATP and adenosine, at the cholinergic neuromuscular junction. I had access to the initial and revised versions of the manuscript, as well as to the comments made by Reviewer 2. In general terms (and in most details), I fully agree with the very important and insightful comments made by the reviewer, which were only partially taken into account by the authors in this revised version.

  1. Review papers should gather the most accurate information possible, giving credit to authors’ original ideas as much as possible. Authors from review papers should refrain from citing other reviews (though more recently published than the original articles) to avoid the risk of hiding authors’ original ideas behind elaborated concepts taken out the appropriate context and without caring about strict experimental conditions used to obtain data. To avoid perpetuating misinterpretations often passed from one review paper to another, careful reading and citation of original articles should be preferred. Thus, the revised manuscript still lacks substantial references to the work of pioneers in the purinergic signaling discovery at the skeletal neuromuscular junction in detriment of more recent self-citations, which is unacceptable in my point of view. The seminal work of Ginsborg, Hirst and Ribeiro, starting in the early seventy’s (past century), followed by the extensive contributions by Silinsky, Sebastião, Smith, Robitaille, Correia-de-Sá and Cunha, in the following two decades until very recently (e.g. doi: 10.1111/jnc.14975), must be adequately acknowledged, as already suggested by the previous reviewer.
  2. One very important concept almost forgotten in the Ziganshin’s review is concerning the pathophysiological implications of adenosine facilitation of nerve-evoked acetylcholine release via the activation of A2A receptors located on rat (and mouse) motor nerve terminals, which has been extensively investigated by the group of Correia-de-Sá. The 1991’s paper from this group (doi: 10.1111/j.1476-5381.1991.tb09836.x) was seminal by revealing that, besides the most studied neuroinhibitory action (in commonly used partially depressed synapses), adenosine could also facilitate the release of acetylcholine through the activation of A2A receptors, depending on calcium and on the nerve firing pattern. This concept was (1) expanded to many other peripheral and central synapses, (2) raised great interest to find novel targets for the treatment of myasthenic syndromes (e.g. doi: 10.1113/jphysiol.2004.067595; doi: 10.1111/j.1471-4159.2011.07216.x; doi: 10.1155/2015/460610), and (3) served as the starting point to the develop the first purine-based antiparkinsonian drug approved recently by FDA.
  3. Another important issue unacceptably disregarded in a present review is concerning the key role of adenosine via A1/A2A receptor balance on the fine-tuning regulation of the neuromuscular transmission by interacting with other presynaptic receptors for acetylcholine (both muscarinic and nicotinic), nitric oxide, neuropeptides (e.g. CGRP, VIP) and trophic factors (e.g. BDNF). The A1/A2A activation equilibrium is highly dependent on the stimulation conditions and on the concentration of the nucleoside at the neuromuscular synapse. Therefore, it is intimately related to the release of ATP (together with acetylcholine) and its subsequent breakdown into adenosine by a cascade of ecto-nucleotidases (see e.g. doi: 10.1113/jphysiol.2003.040410 and doi: 10.1155/2015/460610 in the rat, along with the paper of Cunha & Sebastiao, 1991 in the frog); these include the rate limiting enzyme ecto-5’-nucleotidase/CD73, which exists in close proximity to the A2A receptor in most tissues, including the neuromuscular junction. The mechanisms of adenosine inactivation, both cellular uptake and deamination, are also decisive for the nucleoside neuromodulatory functio, yet no studies available in the literature referring this issue were included in this revision.
  4. By avoiding citation of original papers, authors also did not gave much attention to certain aspects of the experimental conditions where the results were obtained. These include, mixing different animal species (e.g. frogs, mice, rats), paralyzed vs non-paralyzed neuromuscular preparations, low vs high frequency stimulation protocols, among others. Because of their own particular interest, the same strategy was not followed concerning discrimination of ontogenic changes of skeletal neuromuscular junctions and the role of distinct skeletal fiber types, found in self-citation papers. I recommend making a table where data used in the present review is expanded and separated by species and/or other relevant experimental conditions accompanied by reference to the original papers, to avoid gross biases and to increase scientific accuracy.
  5. The pioneering work of the Robitaille’s group about the purinergic signaling role operated by perisynaptic Schwann cells (PSCs) in neuromuscular transmission should be sufficiently emphasized vis a vis their relevance to control acetylcholine signaling and spillover at the motor endplate in the presence of cholinesterase inhibitors or during high frequency tetanic stimuli. This theory was further expanded by Petrov et al (2014) and, more recently, by Noronha-Matos et al (2020), who definitively implicated adenosine released from PSCs, via ENT1, in this mechanism.
  6. Authors emphasized the pathophysiological implications of less known ATP-sensitive P2 purinoceptors, while practically ignoring the putative involvement of much better known adenosine A1 and A2A receptors in highly prevalent neuromuscular diseases, like myasthenia gravis and amyotrophic lateral sclerosis. To be completely fair, an equal treatment should be given to both receptor families, while stressing the limitations of unavailable experimental data.  

Other comments:

  1. The abstract lacks objectiveness not adding much to the current knowledge about the modulatory effect of purines at the neuromuscular junction. It should be rewritten to prompt target audience to read the paper.
  2. It states “In neuromuscular transmission the P2-receptor-mediated influence is minimal under physiological conditions but it becomes very important in some pathophysiological situations such 15 as hypothermia, stress or ischemia.” This hardly emerges as a conclusion from the manuscript due to insufficient experimental data available in the literature to support it.
  3. Line 23, page 1 – “However, embryonic vertebrate skeletal muscle cells initially express receptors for adenosine 5'-triphosphate (ATP), adenosine, glutamate, g-amino butyric acid (GABA) and glycine, as well as for ACh” Adequate references must be provided.
  4. Line 170, page 4 – “It was found that the action of ATP, but not adenosine, is attenuated by the stress hormone cortisol, which allows the efficiency of neuromuscular transmission to be increased sharply under stressful conditions [51-53].” Probably the authors should also comment on the following paper “Amplification of neuromuscular transmission by methylprednisolone involves activation of presynaptic facilitatory adenosine A2A receptors and redistribution of synaptic vesicles” by Oliveira et al (2015, doi: 10.1016/j.neuropharm.2014.09.004).
  5. A theory to explain the paradoxical results described in Table 1 among various types of skeletal muscle fibers in mice and rats should be advanced, after critically reviewing the experimental conditions used to obtain the results.
  6. Figure 2 - This picture requires full explanation. What is the impact of temperature, not only in Ca-induced release, but also on the activity of metabolic enzymes? Drastic changes in temperature are incompatible with life, at least in warm-blood animals (e.g. mice and rats). Please briefly explain the rationale of such experiments in the context of the pathophysiological impact of this review.   
  7. Line 205, page 6 – The interplay between purines and muscarinic receptors function has only been emphasized regarding spontaneous quantal secretion. Nevertheless, there are countless reports in the literature from various research groups demonstrating similar interactions contributing to regulate evoked transmitter release at the neuromuscular junction. Please add this relevant information.
  8. Line 296, page 8 – “Adenosine increases ε-nAChR-channel activity at the end-plate in adult skeletal muscle fibres where the interplay between adenosine and cholinergic systems is mainly mediated by A2B receptors and partially by A3 receptors [54]”. This is from a review paper, please provide the original paper. Are there any imaging studies proving evidence that these two receptors are present in intact skeletal muscle fibres?  

Author Response

The Reviewer 3 comments:

This is a revised version of Ziganshin et al manuscript on the role of purines, mainly ATP and adenosine, at the cholinergic neuromuscular junction. I had access to the initial and revised versions of the manuscript, as well as to the comments made by Reviewer 2. In general terms (and in most details), I fully agree with the very important and insightful comments made by the reviewer, which were only partially taken into account by the authors in this revised version.

Our response:

Thank you for your comment, although we do not fully agree with it since we point-by-point answered all Reviewer’s 2 comments. We included in the revised version of the review the data from about 10 new to the manuscript papers, which had been recommended by the Reviewer 2. In general, we feel that regardless of the unprofessional tone of the Reviewer’s 2 comments we addressed each and every of his/her comments and suggestions.

  1. Review papers should gather the most accurate information possible, giving credit to authors’ original ideas as much as possible. Authors from review papers should refrain from citing other reviews (though more recently published than the original articles) to avoid the risk of hiding authors’ original ideas behind elaborated concepts taken out the appropriate context and without caring about strict experimental conditions used to obtain data. To avoid perpetuating misinterpretations often passed from one review paper to another, careful reading and citation of original articles should be preferred. Thus, the revised manuscript still lacks substantial references to the work of pioneers in the purinergic signaling discovery at the skeletal neuromuscular junction in detriment of more recent self-citations, which is unacceptable in my point of view. The seminal work of Ginsborg, Hirst and Ribeiro, starting in the early seventy’s (past century), followed by the extensive contributions by Silinsky, Sebastião, Smith, Robitaille, Correia-de-Sá and Cunha, in the following two decades until very recently (e.g. doi: 10.1111/jnc.14975), must be adequately acknowledged, as already suggested by the previous reviewer.

Our response:

We absolutely agree with these your comments. However, we did not ignore the pioneering works of the above-mentioned highly respected authors. The main reason why we did not include some seminal papers into our original version of this review was the request of the Editor to keep manuscripts up-to-date – “…50% or above are the papers published within recent 5 years”. However, to meet the Reviewer2’s suggestions, in the revised version of the review we included and acknowledged not only recent, but also some several early papers of Ribeiro, Silinsky, Sebastião, Smith, Robitaille, Correia-de-Sá and Cunha – [14, 15, 16, 17, 18, 23, 30, 31, 32, 36, 44, 45, 49, 50, 51, 53, 56, 63, 89, 98].

  1. One very important concept almost forgotten in the Ziganshin’s review is concerning the pathophysiological implications of adenosine facilitation of nerve-evoked acetylcholine release via the activation of A2A receptors located on rat (and mouse) motor nerve terminals, which has been extensively investigated by the group of Correia-de-Sá. The 1991’s paper from this group (doi: 10.1111/j.1476-5381.1991.tb09836.x) was seminal by revealing that, besides the most studied neuroinhibitory action (in commonly used partially depressed synapses), adenosine could also facilitate the release of acetylcholine through the activation of A2A receptors, depending on calcium and on the nerve firing pattern. This concept was (1) expanded to many other peripheral and central synapses, (2) raised great interest to find novel targets for the treatment of myasthenic syndromes (e.g. doi: 10.1113/jphysiol.2004.067595; doi: 10.1111/j.1471-4159.2011.07216.x; doi: 10.1155/2015/460610), and (3) served as the starting point to the develop the first purine-based antiparkinsonian drug approved recently by FDA.

Our response:

Thank you for this very important comment and suggestion. We mentioned the 1991 paper of Correia-de-Sa et al. [47] when we discussed the mechanism of adenosine action in quantal release of ACh (lines 157-167).  Indeed, we missed the very important pathophysiological role of the concept of adenosine A2A receptor stimulation/inhibition in development of novel drugs for Myasthenia gravis and Parkinson disease. To correct this, we added the following sentence to the manuscript (lines 169-172):

“Stimulation of ACh release by the agonists of adenosine A2A receptors is a promising direction to develop new drugs for treatment of Myasthenia gravis [53] while antagonists of these receptors are considered to be potential anti-parkinsonian drugs [54].”

  1. Another important issue unacceptably disregarded in a present review is concerning the key role of adenosine via A1/A2A receptor balance on the fine-tuning regulation of the neuromuscular transmission by interacting with other presynaptic receptors for acetylcholine (both muscarinic and nicotinic), nitric oxide, neuropeptides (e.g. CGRP, VIP) and trophic factors (e.g. BDNF). The A1/A2A activation equilibrium is highly dependent on the stimulation conditions and on the concentration of the nucleoside at the neuromuscular synapse. Therefore, it is intimately related to the release of ATP (together with acetylcholine) and its subsequent breakdown into adenosine by a cascade of ecto-nucleotidases (see e.g. doi: 10.1113/jphysiol.2003.040410 and doi: 10.1155/2015/460610 in the rat, along with the paper of Cunha & Sebastiao, 1991 in the frog); these include the rate limiting enzyme ecto-5’-nucleotidase/CD73, which exists in close proximity to the A2A receptor in most tissues, including the neuromuscular junction. The mechanisms of adenosine inactivation, both cellular uptake and deamination, are also decisive for the nucleoside neuromodulatory functio, yet no studies available in the literature referring this issue were included in this revision.

Our response:

Thank you for this valuable comment. Indeed, the role of ecto-nucleotidases (mostly CD39 and CD73) was shown to be very important in purinergic transmission in CNS and some peripheral tissues.  Not much data is available about involvement of purine cascade degradation in modulatory roles of purines in skeletal muscles. We mentioned in our review the Cunha & Sebastiao, 1991 paper [16].  In fact, currently we are working on this in our laboratory.

To acknowledge this, we added to our manuscript the following sentence (lines 57-59):

“The regulation of ATP and adenosine degradation is considered to be a novel mechanism to regulate skeletal muscle activity [29] although this needs to be developed further.”

  1. By avoiding citation of original papers, authors also did not gave much attention to certain aspects of the experimental conditions where the results were obtained. These include, mixing different animal species (e.g. frogs, mice, rats), paralyzed vs non-paralyzed neuromuscular preparations, low vs high frequency stimulation protocols, among others. Because of their own particular interest, the same strategy was not followed concerning discrimination of ontogenic changes of skeletal neuromuscular junctions and the role of distinct skeletal fiber types, found in self-citation papers. I recommend making a table where data used in the present review is expanded and separated by species and/or other relevant experimental conditions accompanied by reference to the original papers, to avoid gross biases and to increase scientific accuracy.

Our response:

Thank you for this suggestion. Here again we tried to meet the Editor’s request to include not more than 2-3 Figures/Tables totally. Currently we have 2 Figures and 1 Table.

We made a simple Table of our own data to emphasize different effects of ATP on different types of skeletal muscles.  It would be very interesting to make a Table considering other aspects of experimental protocols, but it is going to be a very complicated table. Developing of such a table takes humongous time, which we could not allow to meet the time-limit for the revision of this paper.

  1. The pioneering work of the Robitaille’s group about the purinergic signaling role operated by perisynaptic Schwann cells (PSCs) in neuromuscular transmission should be sufficiently emphasized vis a vis their relevance to control acetylcholine signaling and spillover at the motor endplate in the presence of cholinesterase inhibitors or during high frequency tetanic stimuli. This theory was further expanded by Petrov et al (2014) and, more recently, by Noronha-Matos et al (2020), who definitively implicated adenosine released from PSCs, via ENT1, in this mechanism.

Our response:

Thank you for this suggestion. We mentioned the Robitaille’s paper in our review under [29]. To emphasize the development of this work we added the following sentence (lines 84-86)

“Recently it was established that α7nAChR controls tetanic-induced ACh spillover from the neuromuscular synapse by promoting adenosine outflow from PSCs via ENT1 transporters and retrograde activation of presynaptic A1 inhibitory receptors [32]”

  1. Authors emphasized the pathophysiological implications of less known ATP-sensitive P2 purinoceptors, while practically ignoring the putative involvement of much better known adenosine A1 and A2A receptors in highly prevalent neuromuscular diseases, like myasthenia gravis and amyotrophic lateral sclerosis. To be completely fair, an equal treatment should be given to both receptor families, while stressing the limitations of unavailable experimental data.

Our response:

Thank you for this comment. Indeed, we tried in our review to focus on less acknowledged ATP-dependent processes rather than on relatively well-known mechanisms and processes for adenosine. However, in this revised version we included some recent data on possible adenosine-involved pathophysiological aspects (please, see above our answers to your comment 2)  

Other comments:

  1. The abstract lacks objectiveness not adding much to the current knowledge about the modulatory effect of purines at the neuromuscular junction. It should be rewritten to prompt target audience to read the paper.

Our response:

We had re-written several sentences in the abstract to add objectiveness (lines 14-17).

  1. It states “In neuromuscular transmission the P2-receptor-mediated influence is minimal under physiological conditions but it becomes very important in some pathophysiological situations such 15 as hypothermia, stress or ischemia.” This hardly emerges as a conclusion from the manuscript due to insufficient experimental data available in the literature to support it.

Our response:

We re-wrote this sentence to be more accurate (lines 14-17)

“As usual, the P2-receptor-mediated influence is minimal under physiological conditions but it becomes very important in some pathophysiological situations such as hypothermia, stress or ischemia. There are some data demonstrating the same in neuromuscular transmission.”

  1. Line 23, page 1 – “However, embryonic vertebrate skeletal muscle cells initially express receptors for adenosine 5'-triphosphate (ATP), adenosine, glutamate, g-amino butyric acid (GABA) and glycine, as well as for ACh” Adequate references must be provided.

Our response:

The reference [4] was added (line 26)

  1. Line 170, page 4 – “It was found that the action of ATP, but not adenosine, is attenuated by the stress hormone cortisol, which allows the efficiency of neuromuscular transmission to be increased sharply under stressful conditions [51-53].” Probably the authors should also comment on the following paper “Amplification of neuromuscular transmission by methylprednisolone involves activation of presynaptic facilitatory adenosine A2A receptors and redistribution of synaptic vesicles” by Oliveira et al (2015, doi: 10.1016/j.neuropharm.2014.09.004).

Our response:

We added the sentence and included the suggested reference (lines 185-187).

“It was also shown that methylprednisolone increases the outflow of ATP from resting motor endplates which may involve activation of presynaptic facilitatory adenosine A2A receptors and synaptic vesicle redistribution [63]”

  1. A theory to explain the paradoxical results described in Table 1 among various types of skeletal muscle fibers in mice and rats should be advanced, after critically reviewing the experimental conditions used to obtain the results.

Our response:

The data included in the Table 1 was obtained in our laboratory using the same experimental conditions. This is why we decided to make a compilation of all those results in a single table.

  1. Figure 2 - This picture requires full explanation. What is the impact of temperature, not only in Ca-induced release, but also on the activity of metabolic enzymes? Drastic changes in temperature are incompatible with life, at least in warm-blood animals (e.g. mice and rats). Please briefly explain the rationale of such experiments in the context of the pathophysiological impact of this review.

Our response:

In Figure 2 we demonstrate the thermo-dependency of synaptic neuromuscular transmission.

Shift of the temperature from 37 to 14oC is not physiological, that is true. But it is not incompatible with life if we are talking about peripheral tissues (like skeletal muscle of the legs and feet), not about the body’s core temperature. In the original papers, which reported the findings of these studies (please, see [68-71]), we described the possible pathophysiological implications of such low temperatures (e.g., induced hypothermia in cardio- and neurosurgery, hibernation, etc.).

  1. Line 205, page 6 – The interplay between purines and muscarinic receptors function has only been emphasized regarding spontaneous quantal secretion. Nevertheless, there are countless reports in the literature from various research groups demonstrating similar interactions contributing to regulate evoked transmitter release at the neuromuscular junction. Please add this relevant information.

Our response:

Indeed, there is a huge number of papers considering the interplay of presynaptic factors regulating induced ACh release. The paragraph is added to touch this big topic (lines 177-180).  

“It was demonstrated that the regulation of ACh release in neuromuscular synapse involves many players on presynaptic level, including purinergic P1 and P2 receptors, muscarinic receptors, as well as receptors for trophic factors [ 56-59].  Interplay of these and may be other factors provides the adaptive changes in the synapse to a given condition [57]”.

  1. Line 296, page 8 – “Adenosine increases ε-nAChR-channel activity at the end-plate in adult skeletal muscle fibres where the interplay between adenosine and cholinergic systems is mainly mediated by A2B receptors and partially by A3 receptors [54]”. This is from a review paper, please provide the original paper.

Our response:

We changed the reference to [80]

Bernareggi, A.; Ren, E.; Giniatullin, A.; Luin, E.; Sciancalepore, M.; Giniatullin, R.; Lorenzon, P. Adenosine promotes endplate nAChR channel activity in adult mouse skeletal muscle fibers via low affinity P1 receptors. Neuroscience 2018, 383, 1-11.

Are there any imaging studies proving evidence that these two receptors are present in intact skeletal muscle fibres? 

Our response:

We are not aware of such studies to be publicly available.

Round 2

Reviewer 1 Report

In the present revised form the manuscript fulfils my requests

Author Response

Thank you for your comments and suggestions.

Reviewer 2 Report

If my aim was to be disrespectful, I would not have written more than one sentence. I only bluntly pointed out that not quoting key studies is disrespectful to all involved and to all working in the field. 

Sadly, the authors were more worried about rebutting the tone than the contents of my comments.

Author Response

Thank you for your comments and suggestions

Reviewer 3 Report

IJMS-868123 v.2 - Modulatory Roles of ATP and Adenosine in Cholinergic Neuromuscular Transmission

This is a re-revised version of Ziganshin et al manuscript on the role of purines, mainly ATP and adenosine, at the cholinergic neuromuscular junction. Authors responded convincingly to most of my criticisms and amended the manuscript accordingly. Despite these improvements, there are still some comments and issues requiring authors’ attention, as follows:   

Issue #1 - Authors state that: The main reason why we did not include some seminal papers into our original version of this review was the request of the Editor to keep manuscripts up-to-date – “…50% or above are the papers published within recent 5 years”. In my opinion this is a very strange requirement under this context. The quality of good scientific results do not decay with time and, therefore, credit must always be given to research pioneers, independently of their discoveries have been repeated since. This perspective is unfair and contribute to deficits in scientific knowledge of newcomers to the field upon reading comprehensive review papers, such the one under evaluation. Nevertheless, I acknowledge authors’ effort to include original papers of research pioneers in studying purinergic signaling at the neuromuscular junction.

Issue #3 – In their answer to the reviewer authors state that “Not much data is available about involvement of purine cascade degradation in modulatory roles of purines in skeletal muscles. We mentioned in our review the Cunha & Sebastiao, 1991 paper [16]. In fact, currently we are working on this in our laboratory.” This is certainly highly acknowledgeable. Yet, authors missed the information detailing the enzymatic kinetics of ATP catabolism and adenosine formation published in the 2003 paper [29] (doi: 10.1113/jphysiol.2003.040410). Stating that “The regulation of ATP and adenosine degradation is considered to be a novel mechanism to regulate skeletal muscle activity [29] although this needs to be developed further” is, therefore, rather short taking into consideration the impact already proven for this pathways in the pathophysiology of neuromuscular transmission (see e.g. doi: 10.1155/2015/460610).

Issue #4 – To avoid further biases and to increase scientific accuracy, my recommendation was that authors should build-up a table where most relevant information published in original papers about the role of purines at the neuromuscular junction appear divided by species, muscle type and other relevant experimental conditions. By doing this authors’ own data contained in Table 1 will become redundant and can be moved to the new table, to comply with editorial requests concerning the number of figures and/or tables. Moreover, time constrains should not refrain authors from presenting good quality data, I am sure editors will take this into consideration.

Author Response

This is a re-revised version of Ziganshin et al manuscript on the role of purines, mainly ATP and adenosine, at the cholinergic neuromuscular junction. Authors responded convincingly to most of my criticisms and amended the manuscript accordingly. Despite these improvements, there are still some comments and issues requiring authors’ attention, as follows:   

Issue #1 - Authors state that: The main reason why we did not include some seminal papers into our original version of this review was the request of the Editor to keep manuscripts up-to-date – “…50% or above are the papers published within recent 5 years”. In my opinion this is a very strange requirement under this context. The quality of good scientific results do not decay with time and, therefore, credit must always be given to research pioneers, independently of their discoveries have been repeated since. This perspective is unfair and contribute to deficits in scientific knowledge of newcomers to the field upon reading comprehensive review papers, such the one under evaluation. Nevertheless, I acknowledge authors’ effort to include original papers of research pioneers in studying purinergic signaling at the neuromuscular junction.

Our response:

Thank you for appreciation of our effort to improve the manuscript. Your valuable comments and suggestions indeed made our review stronger and better.

Issue #3 – In their answer to the reviewer authors state that “Not much data is available about involvement of purine cascade degradation in modulatory roles of purines in skeletal muscles. We mentioned in our review the Cunha & Sebastiao, 1991 paper [16]. In fact, currently we are working on this in our laboratory.” This is certainly highly acknowledgeable. Yet, authors missed the information detailing the enzymatic kinetics of ATP catabolism and adenosine formation published in the 2003 paper [29] (doi: 10.1113/jphysiol.2003.040410). Stating that “The regulation of ATP and adenosine degradation is considered to be a novel mechanism to regulate skeletal muscle activity [29] although this needs to be developed further” is, therefore, rather short taking into consideration the impact already proven for this pathways in the pathophysiology of neuromuscular transmission (see e.g. doi: 10.1155/2015/460610).

Our response:

Thank you for this your comment. Yes, we agree that the pathophysiological aspects are not much considered in our manuscript, since that was out of the main scope of the review. We mostly focused on different physiological aspects of effects of adenosine and ATP in neuromuscular junction. We think that pathophysiological issues of purinergic transmission are well discussed in several recent reviews (doi: 10.1016/j.bcp.2017.07.016; doi: 10.2174/1871527315666160922104848; doi: 10.1007/s11302-013-9381-4)

Issue #4 – To avoid further biases and to increase scientific accuracy, my recommendation was that authors should build-up a table where most relevant information published in original papers about the role of purines at the neuromuscular junction appear divided by species, muscle type and other relevant experimental conditions. By doing this authors’ own data contained in Table 1 will become redundant and can be moved to the new table, to comply with editorial requests concerning the number of figures and/or tables. Moreover, time constrains should not refrain authors from presenting good quality data, I am sure editors will take this into consideration.

Our response:

We understand that the Table 1 and its design are become crucial to our MS. Thus, we completely re-designed the Table 1. We searched the publicly available literature and added data on frogs, guinea pigs and crayfishes into this Table, as well as some new data on rat and mouse skeletal muscles. We constructed the Table 1 in alphabetic order of the names of laboratory animals.  As for experimental conditions of all included studies, we took pragmatic decision not to incorporate these specifics into the Table 1 since almost every study had its own specific experimental protocol (concentration of agents, incubation time, parameters of electrical stimulation, parameters of the recodering, temperature, pH, etc.), none of which was precisely followed by various research teams (authors of cited studies). We demonstrate in the Table 1 current uncertainty in our understanding of purine effects on various skeletal muscle types of various species due to inconsistency of reported effects. For example, we show some contradicting results in rodent (mouse and rat) diaphragm on presynaptic level. There may be multiple reasons for such inconsistencies, including marked variations of experimental designs and protocols. Exploring the measure and direction of the impact of changing of each component of experimental design / protocols deserves a special separate research project. Given that the scope of the current manuscript is to provide a general overview of the effects of adenosine and ATP on neuromuscular transmission, we opted to keep the Table 1 readable and not to overload it with experimental protocol specifics but rather to secure this exciting work for the new future project. We believe that for this manuscript the data presented in the redesigned Table 1 is sufficient. To the best of our knowledge, the Table 1 now consists of all up-to-date data on effects of adenosine and ATP on various skeletal muscles.

Round 3

Reviewer 3 Report

IJMS-868123 v.4 - Modulatory Roles of ATP and Adenosine in Cholinergic Neuromuscular Transmission

The revised version of Ziganshin’s et al. manuscript has been significantly improved, in particular due to review data incorporated in the new Table 1.

Nevertheless, authors should revised carefully references numbering cited in the new parts of the manuscript (including Table 1 and text highlighted in yellow), which do not accurately match those in the references list. This is easily noticed because animal species referred to in Table 1 do not correspond to those of the cited reference.

As per internationally accepted nomenclature, the subtype of all metabotropic receptors (including adenosine and P2Y receptors) should appear in “subscript”. This must be consistent throughout the entire manuscript.

Regarding previous Issue #4, I call authors’ attention that substantial evidence has been provided showing that ATP and adenosine release amounts into the neuromuscular synapse is highly dependent on the stimulus conditions (e.g. phasic vs tetanic) and whether (or not) the neuromuscular safety margin was reduced. As a consequence, purinoceptors tone / activity is influenced by the stimulation pattern used in the experiments. This information should accompany data comparing different species / skeletal muscle types included in Table 1, as it may contribute to elucidate the functional discrepancies emphasized by the authors.

I appreciated the critical analysis of the content of Table 1 made by authors in their response to this reviewer. I suggest this to be incorporated as a final paragraph of this section of the manuscript.  
